# Numerical Simulation of Latent Heat of Solidification for Low Pressure Casting of Aluminum Alloy Wheels

**Qingchun Zheng** [1,2], **Yinhong Xiao** [1,2], **Tao Zhang** [1,2], **Peihao Zhu** [1,2,*], **Wenpeng Ma** [1,2] **and Jingna Liu** [1,2]

[1] Tianjin Key Laboratory for Advanced Mechatronic System Design and Intelligent Control, School of Mechanical Engineering, Tianjin University of Technology, Tianjin 300384, China; zhengqingchun@tjut.edu.cn (Q.Z.); Xiao_Yinhong_tjut@163.com (Y.X.); zhangtao1439@126.com (T.Z.); wenpengma@sina.com (W.M.); liujingna2003@163.com (J.L.)

[2] National Demonstration Center for Experimental Mechanical and Electrical Engineering Education, Tianjin University of Technology, Tianjin 300384, China

[*] Correspondence: zhupeihao_gp@163.com; Tel.: +86-150-2231-0077

**Abstract:** In this paper, aiming at focusses on many problems existing in the mathematical model of temperature change in the low-pressure casting solidification process of aluminum alloy wheel hub, there is a big gap between the simulation and the actual temperature change, which affects the research on the solidification defects of the wheel hub. In order to study the solidification behavior of aluminum alloy hub in low-pressure casting process, the mathematical model describing the temperature change in the process of casting solidification is established by using different solidification latent heat methods. through finite element simulation and experiment, the temperature change in the process of aluminum alloy (A356) solidification is obtained to compare the difference between the temperature change described by different mathematical models, simulation and experiment. The results show that the temperature numerical model of "the temperature compensation heat capacity method" proposed in this paper is most consistent with the simulation temperature change during the solidification process of the aluminum alloy wheel in the simulation mold, which lays a good theoretical foundation for the study of the low-pressure casting process of the aluminum alloy wheel hub.

**Keywords:** low pressure casting; latent heat of solidification; finite element; A356 aluminum alloy

## 1. Introduction

At present, due to the advantages of high rigidity, low density, and good workability of aluminum alloy materials [1], the weight reduction, high strength-to-weight ratio, and good mechanical properties of aluminum alloy wheels have been achieved, which has increased the Automotive applications [2]. These aluminum alloy wheels are usually cast using a low-pressure die-casting process. Low-pressure cast aluminum alloy wheels have the advantages of high production efficiency, low cost, and good mechanical properties. Today, this preparation method has gradually become the main method of wheel manufacturing [3]. However, defects such as shrinkage and porosity often occur during the solidification of the casting. These defects severely affect the mechanical properties of castings, limiting the application of aluminum alloys [4].

There are theoretical difficulties in the research of casting defects. Casting itself is a relatively complicated process. The casting alloy liquid is filled into the cavity at a high temperature for solidification, and accompanied by complex physical and chemical changes, and this process is generally difficult to directly observe. For a long time, the solidification process can only be grasped and controlled through empirical criteria based on a large number of experiments [5]. Therefore, it is

quite difficult to control the quality of castings. During the solidification process of the metal liquid, the latent heat of crystallization is released, and the treatment of the latent heat of the solidification has a great influence on the calculation accuracy of the model [6]. Zhang [7] proposed a non-linear inverse estimation method to accurately estimate the temperature change inside the mold, which is 50% higher than the estimation accuracy of the inversion method. Therefore, in order to improve the accuracy of the numerical simulation of the wheel solidification process, it is of great significance to establish an accurate latent heat model.

In the macroscopic solidification model, there are mainly the sensible heat capacity method, thermal integration method, equivalent heat capacity method, temperature compensation method, enthalpy method, and other latent heat methods applied to different solidification temperature simulation occasions [8]. The sensible heat capacity method is usually used in the case of a wide solid-liquid phase region. For the narrow time step of the solid-liquid region, the equivalent heat capacity method is generally used to improve the accuracy of temperature simulation. In order to solve the problem that the equivalent heat capacity method has a large amount of calculation, the thermal integration method is generally used. However, the accuracy of the heat capacity method depends largely on the time step, and there are still shortcomings [9]. In short, some binary alloys or multicomponent alloys undergo different phase transformations during solidification, and a single latent heat treatment method has greater limitations [10].

In this paper, numerical simulation and theory are combined to study the latent heat numerical model of the solidification process in the low-pressure casting of aluminum alloy wheels. A method of latent solidification heat treatment "the temperature compensation heat capacity method" is proposed. This method combines the temperature recovery method with the equivalent heat capacity method and avoids the system deviation caused by the equivalent heat capacity method when passing through the phase line. It can be applied to alloys with any crystallization temperature range, with a small amount of calculation and high accuracy. The correctness of this method is further studied in this paper. The finite element model is used to simulate the temperature change at a certain point of the wheel hub and the result of this method is compared to verify the accuracy of the method.

## 2. Process and Numerical Methods

### 2.1. Wheel Casting Process

The low-pressure casting process of aluminum alloy wheels is relatively complicated, which mainly includes the processes of die-casting, heat preservation, cooling, and mold removal. The die-casting molds for aluminum alloy wheels are all equipped. A set of molds has several parts: a lower mold, four side molds, an upper mold, a melting furnace, and auxiliary mechanisms [11]. The overall structure is shown in Figure 1.

Usually a die-casting process lasts 3 to 4 min, and the molten aluminum alloy is heated in the furnace. Generally heated to about 700 °C, the aluminum alloy liquid is pressurized by a pressurizer through a lift tube, the aluminum alloy liquid enters the mold cavity through the gate, and is cooled in the cavity. After the aluminum alloy liquid is solidified, the mold is opened, and the mold is opened from the top. The rod mechanism ejects the casting from the mold, thus completing a casting process.

This study mainly studies the temperature change of the cooling process of aluminum alloy liquid in the mold cavity, and the defects of the wheel hub mostly occur in the cooling stage of the aluminum alloy liquid. It is especially important to accurately predict the temperature change of the cooling process in the mold cavity to change the process parameters to reduce the defects of the wheel hub. There are many numerical methods to describe the temperature change during this process. Various methods are different, and the accuracy of description of actual temperature changes is also different. Different numerical models are introduced below. By comparing the numerical simulation data with the finite element simulation data, the accuracy of the methods is obtained.

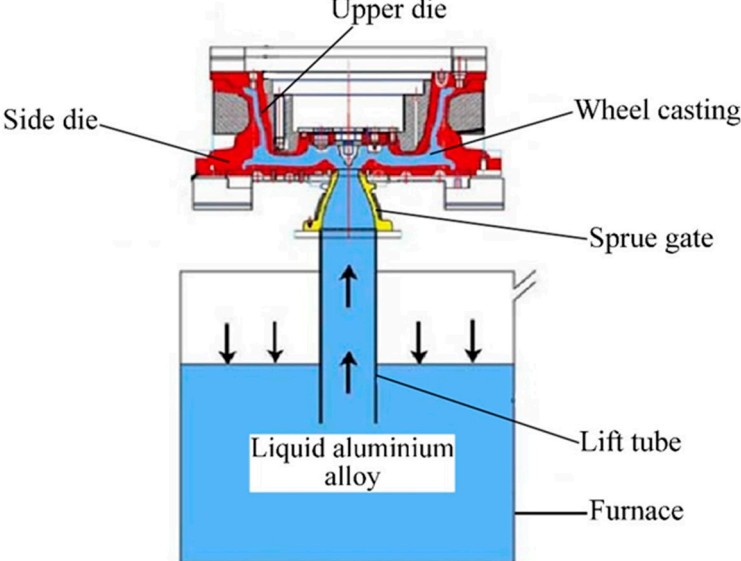

**Figure 1.** Schematic diagram of wheel die casting process.

*2.2. Different Numerical Methods of Latent Heat of Solidification*

At present, latent heat treatment generally adopts temperature compensation method, equivalent heat capacity method, etc. [12]. The temperature compensation method is to increase the temperature of the unit body by the latent heat released by the solidification of the metal. The latent heat released by the solidification of a liquid metal in volume $\Delta V$ is as follows:

$$\Delta Q = \rho L \Delta V \tag{1}$$

If this heat is used to increase its temperature, the temperature should rise:

$$\Delta T = \Delta Q / \rho c \Delta V \tag{2}$$

Although the temperature compensation method seems simple to calculate, it has very obvious shortcomings. Its calculation procedure is to perform a calculation process first, and then perform the next iterative calculation based on the temperature after the rise is obtained. For large cumulative errors, the error will become more and more obvious with the progress of the calculation program, so it is not suitable to use the temperature compensation method to calculate in the entire cooling process.

The equivalent heat capacity method involves the concept of specific heat capacity. The reason why it is called equivalent heat capacity is because in addition to the true specific heat capacity of the alloy material, a part of the specific heat capacity increases $L_0$ due to latent heat is added. In general, in order to use the equivalent heat capacity method, we assume that the latent heat in the molten metal is released uniformly to the outside.

$$L_0 = L / (T_l - T_s) \tag{3}$$

For alloys with a wide crystalline region, this method is more applicable, but for metals or alloys with a narrow solidification region or zero solidification region, a large deviation will occur if it is not comprehensive, where $L_l$ is liquid Line temperature value, $L_s$ is the solid-state line temperature value.

"The temperature compensation heat capacity method" was proposed. Aiming at the various problems mentioned above, based on the existing research, this paper proposes a new solution to the release of latent heat of solidification in the low-pressure casting of aluminum alloy wheels, while using the advantages of the temperature compensation method and the equivalent heat capacity method Using different calculation methods of latent heat of solidification in different cooling intervals, this can not only avoid the cumulative error caused by a large number of iterative operations using a single

temperature compensation method, but also can avoid the defects of the equivalent heat capacity method itself, making the aluminum alloy When the liquid is cooled to the solid-liquid phase line, it does not produce a large error.

According to Fourier's law combined with the first law of thermodynamics, it can be concluded that the amount of heat generated by the heat source within the total heat input equal to the heat accumulation of the total heat transmitted. Because the casting solidification process is designed with multiple equations and energy change processes, the continuity equation, energy equation, and momentum conservation equation must be solved simultaneously. According to the laws of thermodynamics and the differential element method, the differential equation of heat conduction can be derived as [13]:

$$\rho c \frac{\partial T}{\partial t} = \frac{\partial}{\partial x}(\lambda \frac{\partial T}{\partial x}) + \frac{\partial}{\partial y}(\lambda \frac{\partial T}{\partial y}) + \frac{\partial}{\partial z}(\lambda \frac{\partial T}{\partial z}) + q_v \tag{4}$$

Combining with the solidification process modeling method, it can be known that, in order to accurately and reasonably calculate the temperature change of the metal during the solidification process, the effect of the crystallization latent heat release must be added to the metal solidification process. Once the latent heat of crystallization is added, the solidification process of the metal should be regarded as having an internal heat source to heat the molten aluminum, which is different from other metal cooling and heat transfer. At this point, the term in the heat conduction differential equation $q_v$ can be expressed as:

$$q_v = \rho L \frac{\partial f_s}{\partial t} = \rho L \frac{\partial f_s}{\partial T} \cdot \frac{\partial T}{\partial t} \tag{5}$$

$f_s$ is a function of temperature, and the heat conduction differential equation including the release of the latent heat of solidification of the metal is as follows:

$$\rho \left(c - L \frac{\partial f_s}{\partial T}\right) \frac{\partial T}{\partial x^2} = \lambda \left(\frac{\partial^2 T}{\partial x^2} + \frac{\partial^2 T}{\partial y^2} + \frac{\partial^2 T}{\partial z^2}\right) \tag{6}$$

The key of "temperature compensation heat capacity method" to treat the latent heat of solidification is to divide the entire cooling process into different sections, and choose the most suitable method to deal with the cooling and solidifying process in each section. Figure 2 shows four cases when the temperature of the molten metal passes through the phase line. In the figure, $T^n$ is the temperature at the previous moment, $T^{n+1}$ is the temperature at the next moment (without considering latent heat), $\left(T^{n+1}\right)^*$ is the temperature value at the next moment when latent heat is considered, $T_l$ and $T_s$ are the liquidus temperature and the solidus temperature, respectively. Table 1 shows the temperature calculation formulas corresponding to various usage conditions.

**Table 1.** Calculation formula of latent heat model.

| No. | Model Application Conditions | Calculation Formula after Temperature Rise |
|---|---|---|
| 1 | $T^n \geq T_l, T^{n+1} > T_s$ | $\left(T^{n+1}\right)^* = \left[T_l \cdot L/(T_l - T_s) + T^{n+1} \cdot c\right]/\left[c + L/(T_l - T_s)\right]$ |
| 2 | $\left(T_s - T^{n+1}\right)c > (T^n - T_s)L/(T_l - T_s)$ | $\left(T^{n+1}\right)^* = \left[(T^n - T_s)L/(T_l - T_s) + T^{n+1} \cdot c\right]/c$ |
| 3 | $\left(T_s - T^{n+1}\right)c \leq (T^n - T_s)L/(T_l - T_s)$ | $\left(T^{n+1}\right)^* = \left[T^{n+1} \cdot c + T^n \cdot L/(T_l - T_s)\right]/\left[c + L/(T_l - T_s)\right]$ |
| 4 | $\left(T_s - T^{n+1}\right) \cdot c > L$ | $\left(T^{n+1}\right)^* = T^{n+1} \cdot c + L/c$ |
| 5 | $\left(T_s - T^{n+1}\right) \cdot c \leq L$ | $\left(T^{n+1}\right)^* = \left[T_l \cdot L/(T_l - T_s) + T^{n+1} \cdot c\right]/\left[c + L/(T_l - T_s)\right]$ |
| 6 | $T^n \leq T_l, T^{n+1} > T_s$ | $c_e = c + L/(T_l - T_s)$ |

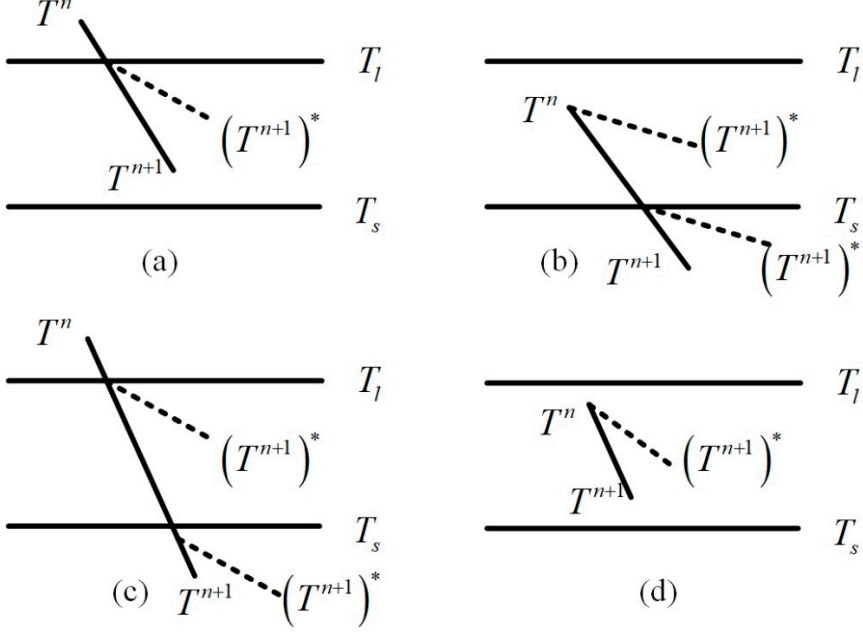

**Figure 2.** Schematic diagram of latent heat treatment: (**a**) When the temperature crosses the liquidus; (**b**) When the temperature crosses the solidus; (**c**) When the temperature of the solid-liquid phase line is close, the temperature at the next moment may be in the solid-liquid coexistence region or in the solid-phase region; (**d**) When the temperature changes in the solid-liquid coexistence region.

### 2.3. Finite Element Analysis

This study uses ProCAST [14] low-pressure casting simulation software to simulate the solidification process of aluminum alloy wheels. ProCAST software is a simulation software for the casting process developed by the United States UES (UNIVERSAL ENERGY SYSTEM) company (Fairfield, CT, USA). The method of numerical calculation and comprehensive solution provides simulation of the casting filling, solidification, and cooling process, and provides many modules and engineering tools to meet the needs of the foundry industry. Based on powerful finite element analysis, it can predict severe distortion, residual stress, temperature, stress, and strain. This study selects a certain point of a wheel hub to generate a solidification temperature change curve, and compares it with a mathematical model of the solidification temperature of the "temperature compensation heat capacity method" to verify its theoretical correctness.

In order to obtain the numerical simulation of the wheel solidification process, several basic steps are included, such as solid modeling of geometric models, division of finite element meshes, setting of material properties and initial conditions, etc. [15].

This study uses UG software to build a three-dimensional model of the wheel hub. UG (Unigraphics NX) is a product engineering solution produced by Siemens PLM Software company (Berlin, Germany). It provides users with digital modeling and verification methods for product design and processing [16]. This study simplifies the original mold assembly structure, leaving the necessary parts for simulating the solidification process, reducing the calculation amount of the simulation process, and analyzing and verifying the simulation results intuitively. The three-dimensional model of the wheel hub and mold is shown in the Figure 3.

After completing the 3D modeling of the wheel hub, the VE module [17] that comes with ProCAST is used to complete the meshing of the model, and the local mesh refinement of some key parts, which guarantees the mesh quality of the model well While minimizing the number of unnecessary grids. Figure 4 shows the meshing of castings and molds.

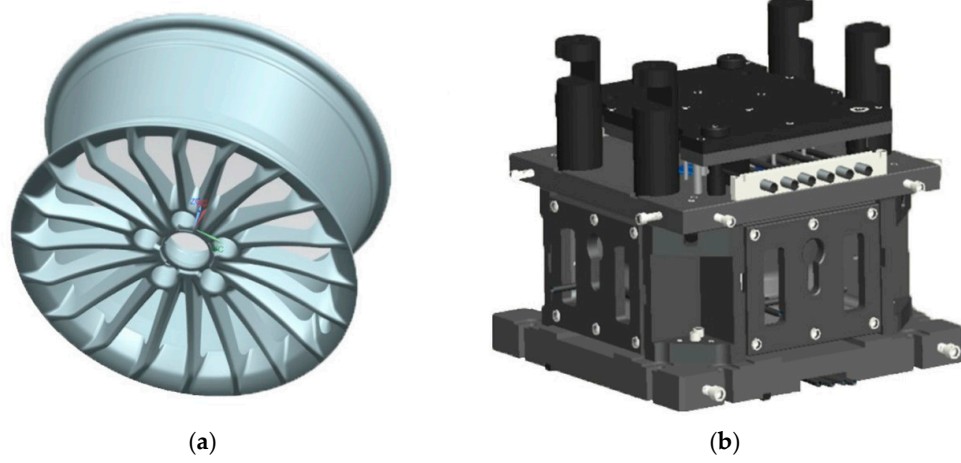

**Figure 3.** Model: (**a**) wheel 3D model; (**b**) mold assembly drawing.

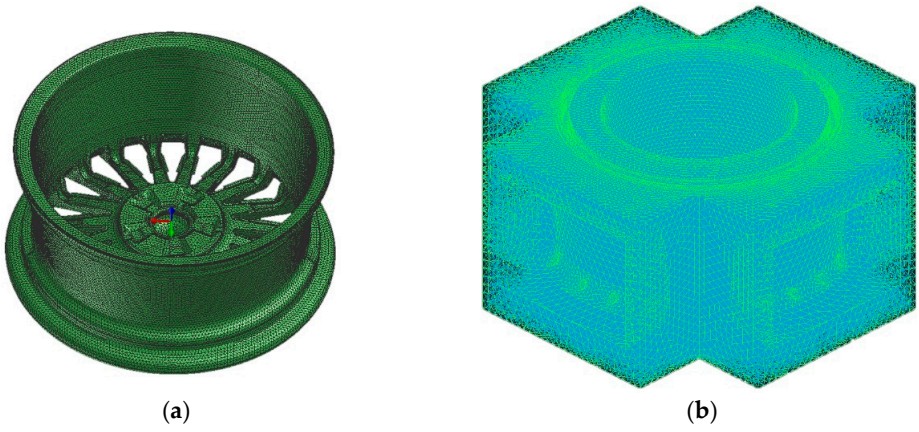

**Figure 4.** Meshing: (**a**) wheel casting meshing; (**b**) die meshing.

The meshed geometric model was imported into the PreCAST module, and the casting materials was defined as A356 aluminum alloy and the die materials was defined as steel-H13. The thermophysical properties of the wheel and die materials used in the model are summarized in Table 2 [18], and the physical properties of the two materials at different temperatures are shown in Tables 3 and 4 [19].

**Table 2.** Thermophysical properties used in the numerical model.

| Material | Properties | Value | Unit |
|---|---|---|---|
| A356 | Thermal conductivity of solid | 70 | W/m/K |
| | Thermal conductivity of liquid | 159.6 | W/m/K |
| | Specific heat | 1150 | J/kg/K |
| | Latent heat | 397,500 | J/kg |
| | Density of solid | 2685 | $Kg/m^3$ |
| | Density of liquid | 2540 | $Kg/m^3$ |
| | Viscosity of the liquid | 0.0014 | Kg/m/s |
| | Partition coefficient of Si | 0.13 | - |
| | Partition coefficient of Mg | 0.48 | - |
| steel-H13 | Density | 7800 | $Kg/m^3$ |
| | Specific heat | 460 | J/kg/K |
| | Thermal conductivity | 24.4 | W/m/K |

**Table 3.** Thermal properties of A356 aluminum alloy at different temperatures.

| Temperature (°C) | 25 | 107 | 200 | 340 | 400 | 579 | 700 |
|---|---|---|---|---|---|---|---|
| Thermal Conductivity [W/(m·k)] | 159.6 | - | 168.2 | - | 229.3 | - | 103.7 |
| Enthalpy (J) | - | $7.3 \times 10^8$ | - | $1.43 \times 10^8$ | - | $1.69 \times 10^8$ | - |

**Table 4.** Thermo-physical parameters of steel material steel-H13 at different temperatures.

| Temperature (°C) | 100 | 200 | 300 | 500 | 700 |
|---|---|---|---|---|---|
| Thermal Conductivity [W/(m·k)] | 20.1 | 20.2 | 22.7 | 23.4 | 24.3 |
| Specific heat [J/(kg·k)] | 468.2 | 525.5 | 560.4 | 612.3 | 685.5 |

According to the effect of on the pressure rate on the filling and the mechanical properties of the wheel hub, the temperature of the preheated mold was set to be 350 °C in this simulation. The initial temperature of mold cavity and gate was set at 700 °C. two parameter values at the gate set: pouring pressure and filling temperature. In general, the rising pressure is 170 mbar to 240 mbar, the filling pressure is 410 mbar to 600 mbar, and the holding pressure is 800 mbar. The pressure curve of the filling process is shown in Figure 5. Set the gate boundary temperature to the pouring temperature, which is 700 °C.

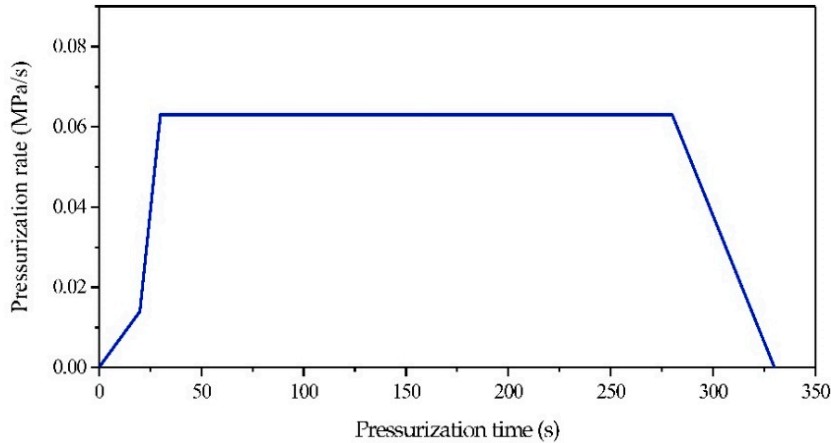

**Figure 5.** Pressure setting curve.

## 3. Plant Trial

In order to obtain the actual solidification latent heat temperature change in the process of wheel hub casting, the factory test was carried out in Tianjin Nanuo Machinery Manufacturing Co., Ltd., Tianjin, China. We select the curing thermocouple at the intersection of hub spoke and cylinder wall. The selection point is shown in Figure 6a below. The selected point is evenly distributed with 4 thermocouples every 90° of the hub. The thermocouples are K-type, with stainless steel sheath and exposed tip to promote good contact with solidified aluminum. The temperature of four thermocouples was recorded at a certain time interval, and the mean value was taken as the temperature of the selected point at that time. The schematic diagram of temperature measurement point A1, A3, and thermocouple installation is shown in the Figure 6b below.

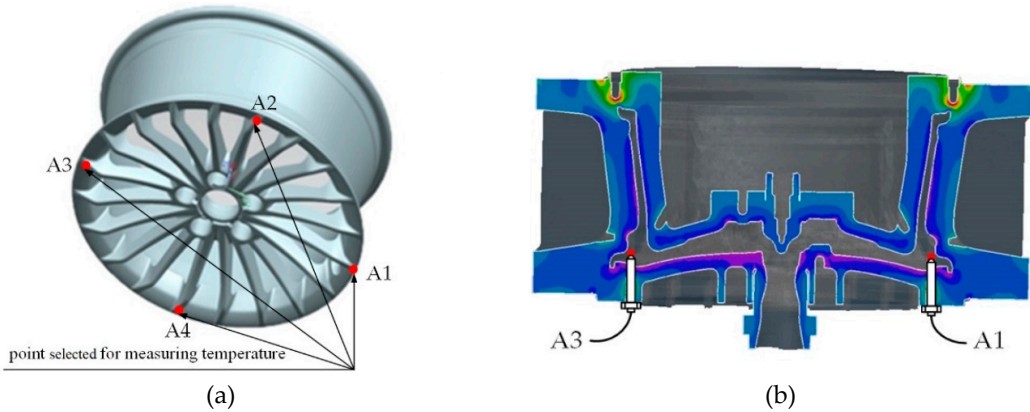

**Figure 6.** Temperature measurement: (**a**) temperature measurement selection point; (**b**) thermocouple installation position.

## 4. Results and Discussion

In the Procast software, the entire low-pressure casting process of the hub was modeled and simulated. The results are shown below. The filling time of the entire wheel is about 20 s and the cooling time is about 300 s. The simulation process of the wheel casting is shown in Figure 7.

Figure 7a–d shows the process of wheel filling, Figure 7e shows the process of wheel filling completed and the begins of cooling and solidification. Figure 7e–h is the cooling and solidification process, and Figure 7h is the solidification process completed. In order to study the temperature change of the solidification process, a point in the casting was selected to perform a finite element simulation of its temperature change.

In the experiment, the temperature is recorded every 5 s in the first 30 s and every 10 s in the last 170 s. The results are shown in the Table 5 below.

**Table 5.** Test point temperature and mean value.

| Sampling Time(s) | A1 (°C) | A2 (°C) | A3 (°C) | A4 (°C) | A Mean (°C) |
|---|---|---|---|---|---|
| 5 | 727 | 732 | 730 | 735 | 731 |
| 10 | 711 | 716 | 715 | 718 | 715 |
| 15 | 700 | 702 | 702 | 704 | 702 |
| 20 | 676 | 680 | 683 | 685 | 681 |
| 25 | 647 | 651 | 650 | 656 | 651 |
| 30 | 621 | 625 | 625 | 629 | 625 |
| 40 | 606 | 607 | 607 | 612 | 608 |
| 50 | 581 | 583 | 588 | 588 | 585 |
| 60 | 578 | 579 | 580 | 583 | 580 |
| 70 | 575 | 574 | 574 | 577 | 575 |
| 80 | 568 | 570 | 572 | 570 | 570 |
| 90 | 559 | 563 | 562 | 564 | 562 |
| 100 | 551 | 553 | 548 | 548 | 550 |
| 110 | 520 | 526 | 526 | 528 | 525 |
| 120 | 483 | 485 | 487 | 485 | 485 |
| 130 | 471 | 473 | 469 | 467 | 470 |
| 140 | 459 | 460 | 456 | 453 | 457 |
| 150 | 447 | 450 | 445 | 446 | 447 |
| 160 | 436 | 440 | 435 | 437 | 437 |
| 170 | 429 | 431 | 426 | 426 | 428 |
| 180 | 421 | 420 | 420 | 423 | 421 |
| 190 | 419 | 417 | 416 | 416 | 417 |
| 200 | 417 | 416 | 414 | 413 | 415 |

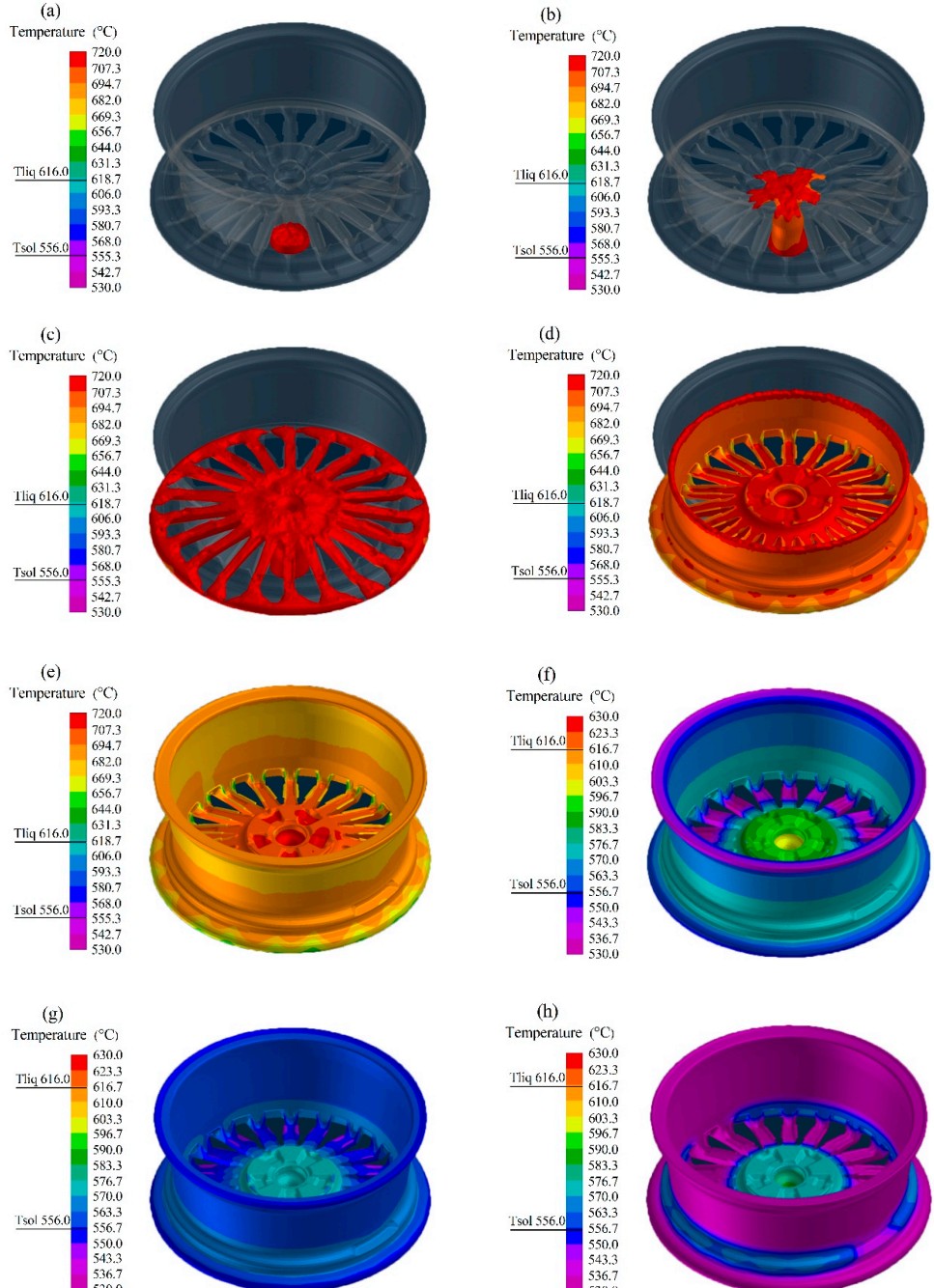

**Figure 7.** Simulation of casting filling and solidification process: (**a**) liquid aluminum alloy rises along lift pipe (**b**) start filling; (**c**) filling is completed by 30%; (**d**) filling is completed by 60%; (**e**) filling completed (**f**) cooling is completed by 30%; (**g**) cooling is completed by 60%; (**h**) cooling completed.

Figure 8 shows three numerical methods A, B, and C. The evolution of hub solidification temperature measured by finite element simulation and experimental results is compared. A Method is the result calculated by temperature compensation heat capacity method, B Method is the result calculated by equivalent heat capacity method, C Method is the result calculated by temperature compensation method, D Simulation is the temperature evolution result obtained by finite element simulation, and E Experiment is the temperature evolution result collected by thermocouple. The results are shown in Figure 8.

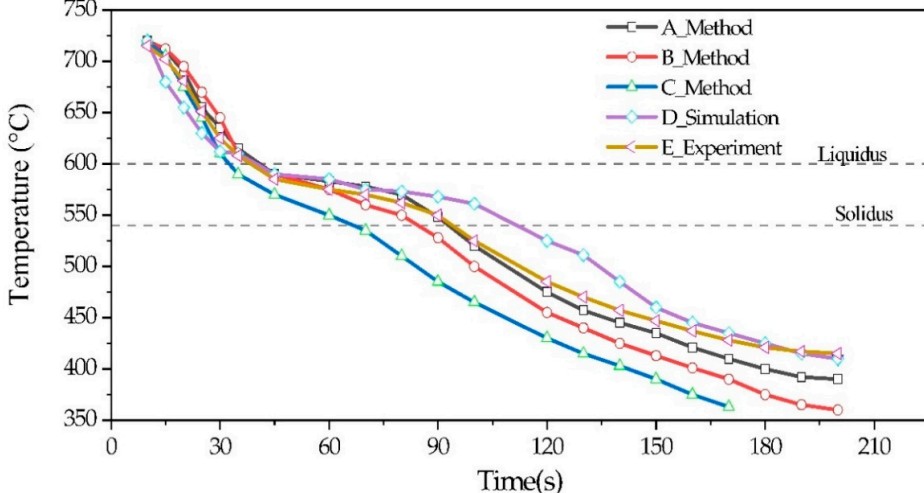

**Figure 8.** Comparison results between different numerical methods of latent heat of solidification and finite element analysis.

Figure 9 shows the temperature error between the A, B, and C numerical methods, simulation, and the experiment value. The 0-reference line represents the same temperature result as the real value. The temperature evolution error of the whole solidification process is shown in Figure 9.

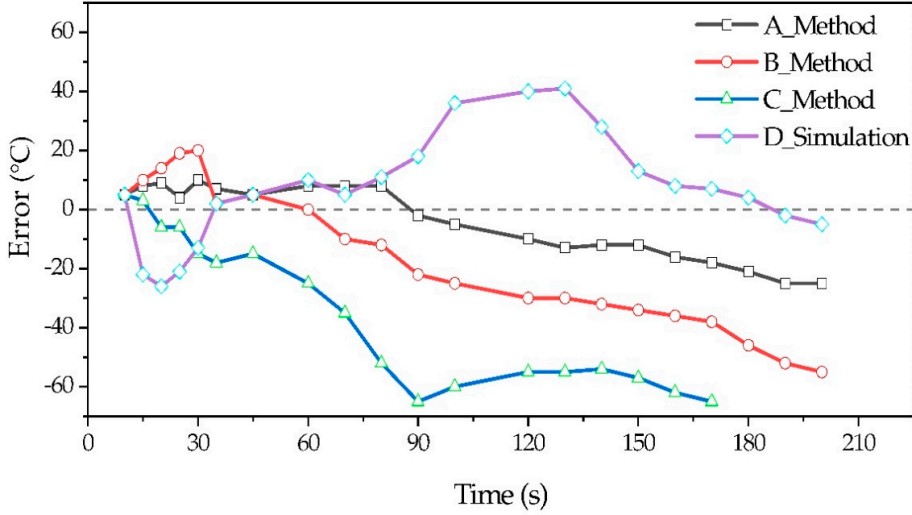

**Figure 9.** The error between the three numerical methods, simulation and the real value.

The influence of latent heat on temperature evolution can be clearly seen from Figure 8. Since the phase transition occurs over a temperature range, there is a transition region from the liquid to the solid phase. In the transition zone, there are both liquid and solid phases, and the temperature change is much gentler than that of single-phase transition temperature. Due to the influence of latent heat of phase transition, part of liquid aluminum alloy is transformed from liquid to solid to release heat, resulting in a gentle temperature change under the coexistence of solid and liquid state. The temperature change of liquid phase is faster than that of solid phase, indicating that the heat release in a single-phase change is closely related to the state of the medium. This paper shows that the heat release of aluminum alloy in liquid phase is faster than that in solid phase.

From the comparison of the calculation results in Figure 8, we can see the shortcomings of numerical methods B and C. When the temperature passes through the solid-liquid phase line, the accuracy calculated by the equivalent heat capacity method will cause a large error, so the calculation accuracy

after 60 s is not high. Although the temperature compensation method does not have the shortcomings of the equivalent heat capacity method, because each calculation is carried out after the iteration, and the temperature after the iteration is not the next temperature value, so the cumulative error will increase with the calculation. The larger accumulation, the less ideal the final calculation result is compared with the actual value. In the calculation process, the temperature compensation method for the casting temperature compensation is very limited, so the temperature drop is the fastest.

The results obtained by the A mathematical model is close to the experimental results. These two curves basically follow the same trend. It is proved that the method of temperature compensation heat capacity method is accurate and reliable. The latent heat calculation method avoids the accumulated error caused by the large amount of iterative calculation of a single temperature compensation method and the defect of the equivalent heat capacity method itself. Therefore, the temperature drop of castings obtained by this method is slower than that obtained by other methods, but it is more reasonable in theory. There is a big error between the temperature change of ProCAST simulation and the real result, especially when the aluminum alloy liquid changes from solid-liquid phase to solid phase, the maximum error is up to 40 °C, which shows that the simulation itself has a big disadvantage, which may be the error between the solidification simulation process and the real one, or the simulation magnifies the influence of latent heat, especially when the solid-liquid phase changes to solid phase. It can be concluded that it is not very accurate to use the finite element simulation method to simulate the temperature change of castings and to discuss the defects of castings during solidification. Figure 9 by comparing the error of temperature values at different times obtained by different calculation methods, the average error of temperature compensation heat capacity method is about 8 °C, and its calculation accuracy is significantly higher than other numerical methods. In conclusion, it is accurate and reliable to use temperature compensation heat capacity method to calculate the solidification process of aluminum alloy containing latent heat.

## 5. Conclusions

In this study, a numerical method for latent heat of solidification for low-pressure casting of aluminum alloy wheels is proposed. By comparing the results of different numerical methods, finite element simulation with the results of experiment, it is found that the "the temperature compensation heat capacity method" has higher accuracy than other numerical methods.

The difference between the calculation and experimental value of the temperature compensation method is the largest, and the error is larger after the temperature passes through the solid phase line due to a large number of iterations. It can be seen from Figure 8 that although the calculation accuracy of the equivalent heat capacity method between the solid-liquid two-phase region is not much different from that of the temperature compensation method, the equivalent heat capacity method is not suitable for processing solid-liquid phase. The temperature change and the curve calculated by the equivalent heat capacity method will obviously deviate from the experimental value when passing through the solid phase line.

The average error between the temperature and the experimental value calculated by the temperature compensation heat capacity method between the aluminum alloy liquid solid-liquid phase line and the experimental value is about 8 °C, and its calculation accuracy is significantly higher than the temperature compensation method and the equivalent heat capacity method. This method is used to describe the temperature change tendency during the low-pressure casting solidification of aluminum alloy wheels with very high accuracy.

**Author Contributions:** Q.Z. was responsible for the drafting of this paper, T.Z. was responsible for the establishment of numerical model, P.Z. was responsible for the theoretical research of numerical model, W.M. performed the experiments, J.L. was responsible for analyzing the data and Y.X. wrote the paper. All authors have read and agreed to the published version of the manuscript.

**Funding:** This research was funded by the National Natural Science Foundation of China (Grant No. 61941305), the Special Funds of Tianjin Municipal Commission of Industry and Information Technology (Grant No. 201803104),



the Tianjin Science and Technology Project (Grant No. 19YFFCYS00110), and the Natural Science Foundation of Tianjin of China (Grant No.18JCQNJC75000).

**Acknowledgments:** The authors gratefully acknowledge the Tianjin Nanuo Machinery Manufacturing Co., LTD for providing the wheel hub equipment and venue used for experiments.

**Conflicts of Interest:** The authors declare no conflict of interest.

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
