# Peer review of "Numerical Simulation of Latent Heat of Solidification for Low Pressure Casting of Aluminum Alloy Wheels"

_metals, doi:10.3390/met10081024_

Round 1

Reviewer 1 Report

In my opinion, the manuscript is interesting and provides interesting information on the possibilities of modeling phenomena occurring in the solidification process. I got the impression that from the text it is not clear at what stage of the simulation the models designated A, B and C were used. It seems to me that the authors should comment more accurately on the differences in obtained results from the adopted numerical models. In my opinion, instead of Fig. 6, showing the degree filling forms for different A, B and C models at the same time would be more useful. I have presented some editorial comments in the attached file. In my opion the article requires minor corrections and after improvement it can be published in Metals issue.

Author Response

Response to Reviewer 1 Comments

Point 1: In my opinion, the manuscript is interesting and provides interesting information on the possibilities of modeling phenomena occurring in the solidification process. I got the impression that from the text it is not clear at what stage of the simulation the models designated A, B and C were used. It seems to me that the authors should comment more accurately on the differences in obtained results from the adopted numerical models. In my opinion, instead of Fig. 6, showing the degree filling forms for different A, B and C models at the same time would be more useful. I have presented some editorial comments in the attached file. In my opion the article requires minor corrections and after improvement it can be published in Metals issue.

Response 1: A, B and C models are different numerical models considering the temperature evolution of latent heat in the solidification stage after the completion of wheel filling. The advantages and disadvantages of different numerical methods have been obtained by comparing and analyzing the errors between the real results and three models. Figure 6 (revised Figure 7) is to simulate the casting process by ProCAST software to obtain the temperature change of solidification process.

Point 2: some comments in the attachment

Response 2:

(1) Figure 2 is not explained. In Figure 2, the explanation of (a) (b) (c) (d) is added, lines 149 to 152, marked in yellow. At the same time, figure 2 is a little fuzzy in the initial manuscripts, and the revised version is replaced by a clear figure.

(2) Wrong expression. Changed to “die materials”. Line 182, marked yellow.

(3) The unit is marked wrong. PA has been deleted. Line 193.

(4) The title of Figure 9 is not appropriate. The title of Figure 9 has been changed to “The error between the three numerical methods, simulation and the real value”. At the same time, the explanation of Fig. 9 is added. Lines 229 to 233, marked yellow.

Reviewer 2 Report

The paper “Numerical Simulation of Latent Heat of Solidification for Low Pressure Casting of Aluminum Alloy Wheels” provides a new numeric model for the calculation of the temperature distribution during solidification of aluminum alloy. The paper is well written. The developed model shows a larger accuracy in comparison with equivalent heat capacity and temperature compensation numerical methods, as well as with finite element simulation. The paper may be accepted for publication after minor corrections accordingly following comments:

  1. The detailed description of the experimental temperature measurement should be added to the Experimental part.
  2. The thermal conductivity of the A356 alloy in Table 1 significantly differs from the value presented in Table 3. What the value was used by the authors for numerical calculation and FEM simulation?
  3. Units for the Error axis should be added in Figure 9.

In line 181 Ref [19] is not correct. Information in Tables 3 and 4 was not given in the paper of Ransing et al. The correct reference should be added.

Author Response

Response to Reviewer 2 Comments

Point 1: The paper “Numerical Simulation of Latent Heat of Solidification for Low Pressure Casting of Aluminum Alloy Wheels” provides a new numeric model for the calculation of the temperature distribution during solidification of aluminum alloy. The paper is well written. The developed model shows a larger accuracy in comparison with equivalent heat capacity and temperature compensation numerical methods, as well as with finite element simulation. The paper may be accepted for publication after minor corrections accordingly following comments:

The detailed description of the experimental temperature measurement should be added to the Experimental part.

Response 1: The experimental description is not detailed. The description of temperature measurement has been added, section 3. Plant trial, lines 198-206, marked yellow.

Point 2: The thermal conductivity of the A356 alloy in Table 1 significantly differs from the value presented in Table 3. What the value was used by the authors for numerical calculation and FEM simulation?

Response 2: Data error in the table. The data in Table 2 and Table 3 are from different references. Table 2 is the thermal conductivity of A356 aluminum alloy liquid at room temperature, and Table 3 is the thermal conductivity at different temperatures. There is an error of 24.6W/m/K in the thermal conductivity at room temperature. my paper calculates the thermal conductivity based on Table 3, and finally decides to change the data in Table 2 to 159.6w /m/K. marked yellow.

Point 3: Units for the Error axis should be added in Figure 9.

Response 3: Figure 9, the vertical axis is not marked with units. Figure 9 has marked units (℃).

Point 4: In line 181 Ref [19] is not correct. Information in Tables 3 and 4 was not given in the paper of Ransing et al. The correct reference should be added.

Response 4: Reference 19 is incorrect. Changed to correct reference. “19. Bakhtiyarov, S.I.;Overfelt, R.A.;Journal of Materials Science 2001, 36, 4643-4648 of Electrical and thermal conductivity of A319 aluminum alloys”. Lines 335 to 336, marked yellow.

Reviewer 3 Report

One simulation or calculation result at one selected point does not appear to appropriate to judge a numerical method. Please use at least 2 experiments and several data points.

Round 2

Reviewer 3 Report

Please use at least to geometries and different data points to support your findings

Author Response

Point 1: Please use at least to geometries and different data points to support your findings
Response 1: Geometric figures and different data points have been added in figure 6 and table 5.
